# Metabolic and Bioprocess Engineering of *Clostridium tyrobutyricum* for Butyl Butyrate Production on Xylose and Shrimp Shell Waste

**DOI:** 10.3390/foods13071009

**Published:** 2024-03-26

**Authors:** Hao Wang, Yingli Chen, Zhihan Yang, Haijun Deng, Yiran Liu, Ping Wei, Zhengming Zhu, Ling Jiang

**Affiliations:** 1College of Biotechnology and Pharmaceutical Engineering, Nanjing Tech University, Nanjing 211816, China; wh@njtech.edu.cn (H.W.); 202161118005@njtech.edu.cn (Y.C.); 202362181226@njtech.edu.cn (Z.Y.); weiping@njtech.edu.cn (P.W.); 2College of Food Science and Light Industry, Nanjing Tech University, Nanjing 211816, China; dhj0858@163.com (H.D.); lyrics7527@163.com (Y.L.); 3State Key Laboratory of Materials-Oriented Chemical Engineering, Nanjing Tech University, Nanjing 211816, China

**Keywords:** agri-food waste, biorefinery, *Clostridium tyrobutyricum*, metabolic engineering, short-chain fatty acid ester

## Abstract

Microbial conversion of agri-food waste to valuable compounds offers a sustainable route to develop the bioeconomy and contribute to sustainable biorefinery. *Clostridium tyrobutyricum* displays a series of native traits suitable for high productivity conversion of agri-food waste, which make it a promising host for the production of various compounds, such as the short-chain fatty acids and their derivative esters products. In this study, a butanol synthetic pathway was constructed in *C. tyrobutyricum*, and then efficient butyl butyrate production through in situ esterification was achieved by the supplementation of lipase into the fermentation. The butyryl-CoA/acyl-CoA transferase (*cat1*) was overexpressed to balance the ratio between precursors butyrate and butanol. Then, a suitable fermentation medium for butyl butyrate production was obtained with xylose as the sole carbon source and shrimp shell waste as the sole nitrogen source. Ultimately, 5.9 g/L of butyl butyrate with a selectivity of 100%, and a productivity of 0.03 g/L·h was achieved under xylose and shrimp shell waste with batch fermentation in a 5 L bioreactor. Transcriptome analyses exhibited an increase in the expression of genes related to the xylose metabolism, nitrogen metabolism, and amino acid metabolism and transport, which reveal the mechanism for the synergistic utilization of xylose and shrimp shell waste. This study presents a novel approach for utilizing xylose and shrimp shell waste to produce butyl butyrate by using an anaerobic fermentative platform based on *C. tyrobutyricum*. This innovative fermentation medium could save the cost of nitrogen sources (~97%) and open up possibilities for converting agri-food waste into other high-value products.

## 1. Introduction

The pre-processing of seafood from fisheries and catering industries produces diverse waste and by-products. Annually, the crab and shrimp processing industrials in the EU generate over 100,000 MT of shellfish waste [1]. Furthermore, lobster processing results in the production of 50–70% shellfish by-products, including shells, heads, eggs, and livers, resulting in an annual total loss exceeding 50,000 MT [2]. However, these discards contain various valuable components, such as amino acids, protein, lipids, chitin, calcium carbonate and other macro/micro elements [3]. Currently, the biological treatments of shrimp shell waste like bio-catalysis and biotransformation are widely investigated [4,5,6], which enables a simple, efficient, and economic bioprocess for the utilization of shrimp shell waste. It was found that the dry weight of shrimp shell consists of 18% chitin, 43% protein, 29% ash and 10% lipid, indicating that shrimp shell waste is rich in protein, which is a high-quality nitrogen source for microbial growth [7]. Exploring natural, stable and suitable alternative nitrogen sources has become one of the limiting factors for the development of the fermentation industry [8].

Biomanufacturing, especially microbial fermentation, can transform agri-food waste into various chemicals and biofuels, which offers a sustainable and cleaner alternative [9]. Achieving efficient use of agri-food waste has always been one of the goals of microbial fermentation. *Clostridium* sp., a group of anaerobic bacteria, can produce various high-value products from three generations of feedstocks, such as biomass crops, lignocellulosic biomass, and C1 gases (i.e., CO and CO_2_). Therefore, *Clostridia* species were considered promising hosts for biorefineries, especially from agri-food waste [10]. Butyrate is a valuable product that can be produced by *Clostridium tyrobutyricum*, a member of the *Clostridia* group [11]. *C. tyrobutyricum* strains can metabolize various carbon sources, such as glucose, fructose, and mannitol, to produce butyrate with high titer, yield, and productivity [12]. However, the economic feasibility of butyrate production depends on the availability and cost of the feedstocks. Therefore, several renewable and low-cost feedstocks, such as glycerol, macroalgal biomass, food waste, and lignocellulosic biomass, have been investigated for their potential to support butyrate production by *C. tyrobutyricum* [13,14]. As the main carbon source released from the lignocellulosic biomass, xylose can be consumed by *C. tyrobutyricum* to achieve efficient fermentation [15].

As one of the short-chain fatty acid esters (SCFAEs), butyl butyrate has various applications in different industries, especially in foods [16]. It has a distinctive fruity aroma that resembles pineapple and banana, which makes it a desirable food additive to improve flavor. So, there is a growing interest in developing a safe and efficient microorganism as a cell factory for the production of bio-based butyl butyrate. Recently, a microbial co-culture system was developed, which consists of butyrate- and butanol-producing strains; then, butyrate and butanol are condensed to form butyl butyrate. As a result, 3.0 g/L of butyl butyrate can be achieved from the co-fermentation system [17]. In addition, as a potential next-generation probiotic, *C. tyrobutyricum* has the main cell factory for butyl butyrate production, which can be achieved by either de novo synthesis or bio-catalysis approaches [18,19]. One-step direct fermentation of fermentable sugars by *C. tyrobutyricum* is the main method for butyl butyrate production. This method involves the in vivo condensation of butyryl-CoA and butanol by heterologous expression of alcohol acyltransferase (AAT), without the addition of any precursors or lipases [16]. For example, using efficient, robust genome-editing strategies, the engineering strain could produce 62.6 g/L of butyl butyrate with a selectivity of 96.97% from mannitol [18,20]. However, AAT has a broad substrate specificity, and the introduction of the *AAT* gene also leads to the synthesis of ethyl butyrate [21]. Therefore, it is necessary to explore AAT enzymes that have higher substrate specificity for butyl butyrate synthesis. In addition, plant-derived AAT enzymes still have problems such as low affinity and catalytic activity [22]. In addition to this, another ideal approach for butyl butyrate synthesis is condensing fermented butyrate and butanol using lipases as catalysts [23].

To achieve low-cost and high-selectivity production of butyl butyrate, we developed a promising fermentation medium for butyl butyrate production with xylose as the sole carbon source and shrimp shell waste as the sole nitrogen source. Alongside the fermentation medium, the butanol synthetic pathway was introduced into *C. tyrobutyricum*, and then butyl butyrate production was achieved through in situ esterification by the supplementation of lipase into the fermentation. Furthermore, the butyryl-CoA/acyl-CoA transferase (*cat1*) was overexpressed to balance the ratio between precursors butyrate and butanol. Finally, the low-cost production of butyl butyrate was achieved, without the generation of the byproduct ethyl butyrate. Additionally, the surprising outcome of the enhanced xylose utilization was further investigated during the fermentation by transcriptome analysis. Overall, we paved a promising route for high-valued SCFAEs production by using xylose and shrimp shell waste, which might also accelerate the understanding of the synergistic utilization of xylose and shrimp shell waste in *C. tyrobutyricum*.

## 2. Materials and Methods

### 2.1. Bacterial Strains and Growth Conditions

The plasmids and strains used in this study are listed in Table 1. The *C. tyrobutyricum* L319 strain was isolated and stored in our previous study [24]. The *Escherichia coli* JM109 strain was used for genetic experiments and the *E. coli* CA434 strain was used to deliver the plasmid into *C. tyrobutyricum* L319 cells [25]. For *C. tyrobutyricum* cultivation, Reinforced Clostridial Medium (RCM) [26] was used for seed cultivation, and Clostridial Growth Medium (CGM) was used for fermentation experiments, which contained (per liter) 5 g tryptone, 5 g yeast extract, 5 g NaCl, 3 g (NH_4_)_2_SO_4_, 1.5 g K_2_HPO_4_, 0.6 g MgSO_4_·7H_2_O, 0.03 g FeSO_4_·7H_2_O, and specific carbon sources. In addition, *E. coli* was cultivated in a Luria–Bertani (LB) medium (10 g/L tryptone, 5 g/L yeast extract, 10 g/L NaCl). 

### 2.2. Development of the Shrimp Shell Powder Medium (SSP Medium)

The shrimp shells (mainly from *Oratosquilla oratoria*) used in the culture medium were collected from kitchen waste, subjected to cleaning, drying, crushing, and grinding into powder. They were then filtered through a 100-mesh sieve to obtain shrimp shell powder suitable for use in fermentation experiments. These shrimp shell powders served as the nitrogen source in the culture medium. Furthermore, the inorganic salt ions in the culture medium were formulated based on the design of CGM, with the choice of the carbon source being contingent on specific conditions.

### 2.3. In Situ Catalytic Synthesis of Butyl Butyrate

In the fermentation process, the catalytic synthesis of butyl butyrate was performed by using lipase Novozyme 435. The growing seed culture was inoculated (at 10% *V*/*V*) into the serum bottle and the lipase was added with the hexadecane to the fermentation system early in the log phase (~12 h), keeping the ratio of the aqueous phase and organic phase at 2:1 (*V*/*V*). The catalytic synthesis was carried out at 150 rpm and 37 °C. The organic acids and alcohol substances produced during fermentation were predominantly present in the aqueous phase due to partition coefficient factors [23]. However, the ester products generated through catalysis were extracted into the organic phase (hexadecane was used in the study). Therefore, the concentration of organic acids and sugars was assessed by analyzing the aqueous phase, while the ester titer was determined in hexadecane [23].

### 2.4. Construction of the Recombinant Strain

The plasmids used are listed in Table 1. For the heterologous expression of the aldehyde/alcohol dehydrogenase (*adhE2*) in *C. tyrobutyricum*, it was isolated and amplified from the genome of *C. acetobutylicum* ATCC824 using primers *adhE2*-F/*adhE2*-R, and the *thl* (GTH52_RS11340) promoter was amplified from *C. tyrobutyricum* L319 genome DNA using primers p*thl*-F/p*thl*-R. The shuttle vector pMTL82151 was used for the heterologous expression of *adhE2* from *C. acetobutylicum* ATCC824. Subsequently, they were cloned into the linearized vector pMTL82151 using primers V*adhE2*-F/V*pthl*-R, finally generating the recombinant plasmid pMTL-P*thl*-*adhE2* [27]. Likewise, the endogenous overexpression of the butyryl-CoA/acetate CoA transferase (*cat1*) (GTH52_RS08875) was also carried out based on the plasmid pMTL-P*thl*-*adhE2*. The gene *cat1* and its native promoter were amplified from *C. tyrobutyricum* L319 genome DNA using primers *cat1*-FOR/*cat1*-REV, and then cloned into the linearized vector amplified from pMTL-P*thl*-*adhE2* using primers V*cat1*-F/V*cat1*-R*,* generating the recombinant plasmid pMTL-P*thl-adhE2*::P*cat1*-*cat1*. All the primers used in the study are listed in Appendix A.

Upon being constructed, the recombinant plasmids were conjugated into *C. tyrobutyricum* L319 cells via the *E. coli* CA434 strain [28]. Colony PCR was used to confirm the successful transformation.

### 2.5. Optimization of Carbon Sources in SSP Medium

Further screening of the optimized carbon sources in the fermentation was conducted in the serum bottle based upon the SSP medium at 37 °C in static cultivation anaerobically. The choice of carbon sources was guided by prior experimental discoveries [29]. When using a single carbon source, options included glucose, xylose, chitooligosaccharides, and galactooligosaccharides. For experiments involving mixed carbon sources, the carbon mole (C-mole) of different carbon sources was maintained to ensure the accuracy of experimental outcomes. The calculation equation is as follows: C-mole = *c* (concentration)/M (molar mass) × N (number of carbon atoms).

### 2.6. Optimization of the pH Values during Fermentation

Fermentation experiments were conducted with optimized media at varying pH levels in the serum bottle. The selection of pH values, specifically 5.5, 6.0, and 6.5, was guided by previous observations of the growth of *C. tyrobutyricum* L319 under different pH conditions. pH adjustment was achieved using 3 M hydrochloric acid. Samples were collected at different growth stages for growth and product curves, with at least three replicates for each condition.

### 2.7. The Scale-Up Fermentation in a 5 L Bioreactor

In order to achieve a high titer of butyl butyrate during fermentation using xylose and SSP medium, a 5 L bioreactor (T&J Bioengineering Co., Ltd., Shanghai, China) was used at 100 rpm agitation at 37 °C. Before inoculating the strain, the nitrogen was pumped into the broth for 30 min to make it anaerobic. During the fermentation, samples were collected at different growth stages.

### 2.8. Analytical Methods

A conventional spectrophotometer (Shanghai Spectrum, Shanghai, China) was employed to determine cell density via optical density measurement at a wavelength of 600 nm (OD_600_).

Glucose, acids, and alcohol concentrations were determined using high-performance liquid chromatography (HPLC) with an LC-15C instrument (Shimadzu, Kyoto, Japan), which was equipped with a refractive index detector (RID) and an Aminex HPX-87H ion exclusion column (Bio-Rad, Hercules, CA, USA) maintained at 65 °C. At a flow rate of 0.6 mL/min, 5 mM H_2_SO_4_ was used as a mobile phase [30].

The butyl butyrate in the organic phase was assessed using gas chromatography (GC2010, Shimadzu, Japan) featuring a flame ionization detector (FID). A Restek-1 capillary column (30 m × 0.25 mm × 0.5 μm, Restek Corporation, Bellefonte, PA, USA) was utilized with N_2_ as the carrier gas at a flow rate of 30 mL/min. The GC oven temperature program was programmed as follows: initial temperature of 180 °C held for 2 min, ramping up at 5 °C/min to 200 °C, further ramping up at 30 °C/min to 250 °C, followed by a 5 min hold. A 1 μL sample was injected using the split mode (30:1). The injector and detector temperatures were maintained at 250 °C and 280 °C, respectively.

### 2.9. Statistical Analysis

The obtained data were statistically analyzed using one-way analysis of variance (ANOVA) and expressed as mean ± standard error. The correlation coefficients of the mentioned parameters were analyzed using GraphPad Prism (GraphPad Prism 9.5.1, Boston, MA, USA, www.graphpad.com, accessed on 1 May 2023).

### 2.10. Transcriptome Analysis

Samples were collected at the mid-log phase and stored in liquid nitrogen until being sent to Chengwen Biotechnology Co., Ltd. (Nanjing, China) for the transcriptome analysis. Total RNA was extracted using TruSeq^TM^ Stranded Total RNA Library Prep Kit. The assessment of RNA quality was carried out using Nanodrop2000 while the DNA’s integrity was evaluated via standard 1% agarose gel electrophoresis. RNA sequencing was executed using an Illumina MiSeq 250 Sequencer (Illumina, San Diego, CA, USA). Subsequently, bioinformatic analysis was conducted with chimera checking, which involved comparison with the Gold database (http://drive5.com/uchime/gold.fa, accessed on 1 May 2023). Redundancy analysis (RDA) was conducted using the software Canoco for Windows 4.5 (Biometris, Wageningen, The Netherlands) and assessed through Monte Carlo permutation procedures with 499 random permutations. All statistical analyses were performed using SPSS 22.0 for Windows (SPSS Inc., Chicago, IL, USA). Resulting *p*-values were adjusted for multiple testing using the Benjamini–Hochberg false discovery rate (FDR) correction. Only FDR-corrected *p*-values below 0.05 were considered statistically significant. The experiments were conducted in triplicate.

## 3. Results

### 3.1. Establishment of the Biosynthetic Pathway for Butyl Butyrate Production in C. tyrobutyricum

The hyper-butyrate-producing strain, *C. tyrobutyricum* L319, was used to produce butanol by introducing the bifunctional aldehyde/alcohol dehydrogenase gene *adhE2* through the generation of the recombinant strain CtAD. (Figure 1A,B) (Table 1). Based on the synthetic pathway of butyrate and butanol, efficient butyl butyrate production through in situ esterification was achieved by the supplementation of lipase (Novozyme 435) into the fermentation (Figure 1A). As expected, after the overexpression of *adhE2* under the *thl* promoter, the recombinant strain could produce 0.7 g/L of butyrate and 3.3 g/L of butanol, and the butyl butyrate production of 1.4 g/L was achieved in a fermentation (CGM medium) through the catalysis of esterification with lipase for 12 h to the maximum titer (Figure 1C). These results revealed that the biosynthetic pathway of butyl butyrate was successfully established in *C. tyrobutyricum* by combining metabolic engineering and in situ esterification strategies.

### 3.2. Efficient Butyl Butyrate Production from Shrimp Shell Waste

The shrimp shell powder (SSP) and ion components were sequentially added to the medium for determining the growth performance of the strain CtAD. It was found that the strain CtAD was unable to grow in a medium only containing glucose and SSP (Figure 1D). This might have been due to the limited residual nutrients in the SSP, which lacked the essential growth factors required by the strain, such as metal ions. So, the optimization of CGM was further conducted by replacing the nitrogen sources (i.e., tryptone, yeast extract and (NH_4_)_2_SO_4_) with SSP while retaining the ion compositions; this modified medium was finally defined as the SSP medium (per liter, 25 g shrimp shell powder, 5 g NaCl, 1.5 g K_2_HPO_4_, 0.6 g MgSO_4_·7H_2_O, 0.03 g FeSO_4_·7H_2_O). Interestingly, the strain CtAD could grow normally when fed with glucose in the SSP medium (Figure 1D). Additionally, CtAD exhibited similar growth performances in the SSP medium as in CGM. However, the lag phase of CtAD became slightly longer in the SSP medium, which was primarily due to fact that the SSP could not be directly utilized by CtAD as a nitrogen source (Figure 1E). Consequently, before being utilized, the SSP ought to be first hydrolyzed by proteases. However, this process is also limited by the initial vitality and quantity of the strain.

The fermentation products of the CtAD strain in the SSP medium were also investigated; it was found that the titer of butanol was rapidly decreased by 35%, and the acetate was also slightly decreased, which might have been due to the buffering agent in the SSP medium that controlled the pH around 6.0. The stable pH is more preferable for butyrate production than acetate; the titer of butyrate was increased by 60% while the titer of butanol was decreased. Finally, the titer of the butyl butyrate was increased by 8%, and 1.52 g/L of butyl butyrate was produced (Figure 1C). So, SSP could be used as a promising feedstock for butyl butyrate production.

### 3.3. Preliminary Economic Evaluation of Fermentation on SSP Medium

The production cost depends on the choice of nitrogen and carbon sources during the fermentation. Therefore, the cost of nitrogen sources in two media were compared; these were the SSP medium, which uses the remaining protein in the shrimp shell as the nitrogen source, and the CGM medium, which uses commercial nitrogen-source compounds (tryptone, yeast extract and (NH_4_)_2_SO_4_). The SSP medium has a great economic and environmental advantage because it utilizes the shrimp shell powder, which is derived from the abundant and cheap shellfish waste (5.57 million tons annually) [31]. The cost of nitrogen sources per liter was $0.0025–0.0030 (25 g/L) for the SSP medium while it was $0.0780 (13 g/L) for the CGM medium (Table 2). The price of the commercial compounds was filtered in the Import Genius (https://www.importgenius.com/, accessed on 1 November 2023) search engine according to the HS code of each compound. Moreover, the SSP medium increased the final titer of butyl butyrate and reduced the production cost by 97% ($0.0016–0.0019 vs. $0.0560 per gram). So, the SSP medium was an effective and low-cost method for butyl butyrate fermentation.

### 3.4. Balancing the Ratio between Butyrate and Butanol

To achieve the efficient production of butyl butyrate, it is important to keep the molar ratio of butyrate and butanol to 1. The imbalance of the ratio between butyrate and butanol leads to the limit of the enhanced synthesis of butyl butyrate (Figure 1C). To balance the ratio between butyrate and butanol, the gene *cat1* was overexpressed under the *cat1* promoter, generating the strain CtADGBC (Figure 2A,B). Acetate in the strain CtADGBC was decreased by 36% as compared to the strain CtAD (2.3 g/L vs. 3.6 g/L). In addition, butyrate in the strain CtADGBC (2.2 g/L) was increased by 90% as compared to the strain CtAD (1.2 g/L). However, the strain CtADGBC exhibited a reduction in butanol production (2.1 g/L vs. 1.5 g/L). Thus, the different titer between butyrate and butanol was reduced (0.5 vs. 1.6), which was more favorable for the synthesis of butyl butyrate (Figure 2C). All these results indicated that the overexpression of *cat1* could ultimately balance the ratio between butyrate and butanol when fed with glucose.

### 3.5. Enhanced Butyl Butyrate Production by Using Xylose as the Sole Carbon Source in the SSP Medium

The optimization of carbon sources, both single and mixed, was investigated. For glucose (30 g/L) and xylose (30 g/L), their high purity enabled the calculation of corresponding concentrations using the molar amount of the carbon formula. However, for chitooligosaccharide (3 g/L) and galactooligosaccharide (5 g/L), which are mixtures, precise determination via the formula was not feasible. Therefore, we calculated the corresponding concentrations according to the molar mass multiples. Finally, it could be observed that, in the SSP medium, the strain CtADGBC could grow on both glucose and xylose (Figure 3A). Moreover, a shorter lag phase (5 h vs. 8 h) and a higher biomass (OD_600max_ 12 vs. 10) were exhibited compared to those in the xylose-fed medium. Despite the higher biomass being found in the glucose-fed medium, the byproduct acetate was the main metabolite obtained from the glucose, followed by butyrate and butanol; this did not contribute to butyl butyrate synthesis. Nevertheless, butyrate (4.5 g/L) and butanol (2.5 g/L) achieved the highest titer by using xylose as the sole carbon source, compared to the results with glucose as the sole carbon source (Figure 3B). As a result, 2.3 g/L of butyl butyrate was obtained, leading to an increase of 53% as compared to the results with glucose as the sole carbon source (Figure 3C). As a matter of fact, the titers of butyrate and butanol were both increased and the balance of their ratio (1.8 vs. 2.6) was further improved when fed with xylose, thus boosting the production of butyl butyrate.

### 3.6. Impact of Acidic Conditions on the Production of Butyl Butyrate

To investigate whether the pH affects the production of butyl butyrate in the SSP medium when fed with xylose, the pH values of the medium were adjusted to 5.5, 6.0 and 6.5. Despite the growth performance of CtADGBC in different pH values being quite similar (Figure 4A), the titer of acetate was decreased by 46% and the butyrate was increased by 23% at pH 6.0 compared with that at pH 5.5 (Figure 4B). Meanwhile, the titer of butanol became higher when the pH rose. Overall, the most suitable pH value for the production of butyl butyrate in the SSP medium (with xylose) was pH 6.0, and 2.5 g/L of butyl butyrate was generated. If not otherwise mentioned, the pH was set at 6.0 throughout the following experiments.

### 3.7. Efficient Butyl Butyrate Production in a 5 L Bioreactor

The scale-up batch fermentation in a 5 L bioreactor was performed to boost the efficient production of butyl butyrate. Since the buffering complexes are naturally contained in the SSP, such as CaCO_3_, the pH during the fermentation could be auto-controlled around 6.0 in the SSP medium. Compared with the fermentation in serum bottles, the cell density (OD_600_) increased by 2-fold (20 vs. 10) (Figure 5A). Moreover, the titer of butyrate achieved 7.1 g/L, resulting in a 26.8% improvement as compared to that in a serum bottle (Figure 5B). Furthermore, 70 g/L of xylose was consumed within 55 h (Figure 5C). The xylose consumption was rapid and reached a rate of 1.3 g/L·h, which had never been achieved before with *C. tyrobutyricum* feeding on xylose as its sole carbon source. Meanwhile, the concentration of butyrate and butanol reached the peak when 100% xylose was consumed, which indicated that the generation of the precursors (butyrate and butanol) could not keep pace with the synthesis of butyl butyrate (Figure 5B). Moreover, the concentration of acetate remained almost constant, indicating that insufficient reducing power might inhibit the transformation of acetate to butyrate (Figure 5B). Overall, 5.9 g/L of butyl butyrate was finally achieved with a selectivity of 100% and a productivity of 0.03 g/L·h (Figure 5D).

### 3.8. Transcriptome Analysis Probing Relevant Mechanisms Associated with Xylose and SSP Utilization in the SSP Medium

During fermentation in the SSP medium, xylose consumption was significantly enhanced. The xylose consumption rates in the CGM and SSP medium were 0.2 g/L·h and 1.1 g/L·h, respectively, when fed with 50 g/L of xylose (Figure 6). In order to further investigate the effects of xylose and SSP on the metabolism of the CtADGBC in the SSP medium, differences in gene transcription between CGM and the SSP medium were analyzed using transcriptomics. Comparative transcriptome data showed that 1351 differentially expressed genes (DEGs), including 676 up-regulated genes and 675 down-regulated genes, were observed in CtADGBC when cultured in the SSP medium (Appendix A). The KEGG annotation analysis revealed that the difference in genetic transcription was found mainly in the nitrogen metabolism, carbohydrate metabolism and amino acid metabolism and transport (Appendix A).

It could be found that the genes encoding xylose isomerase (*xylA*) and xylulokinase (*xylB*) were found up-regulated by 2.6- and 3.2-fold, respectively, in the SSP medium compared that CGM. These two genes were mainly involved in the xylose metabolism, which indicated that the strain CtADGBC improved the utilization of xylose by up-regulating relevant genes in the xylose metabolic pathway. Furthermore, the expression levels of the key genes involved in the nitrogen metabolism, including *nifD*, *nifK*, and *nifH*, increased significantly in the SSP medium. These genes could contributed to the synthesis of ammonia. Meanwhile, several genes implicated in glutamate synthesis were up-regulated. Among them, the *glnA*, *gltB*, and *gltD* genes showed dramatic 5.0-, 7.1-, and 8.9-fold up-regulations, respectively. We also observed significant changes in the expression of the genes involved in the aspartate and methionine metabolism. Importantly, the genes involved in amino acid transport, including GTH52_RS08220, *glnP, glnH*, and *glnQ* showed dramatic 128.7-, 117.0-, 41.7-, and 36.1-fold up-regulation, respectively. All the results indicated that the strain CtADGBC improved the utilization of SSP by up-regulating the relevant genes involved in ammonia synthesis, amino acid metabolism and transport (Figure 7) (Appendix A).

## 4. Discussion

In this study, a butanol synthetic pathway was introduced into *C. tyrobutyricum* L319, and then efficient butyl butyrate production through in situ esterification was achieved by the supplementation of lipase into the fermentation. As the precursors of in situ esterification, it is important to keep the molar ratio of butyrate and butanol to 1; an imbalance in the ratio between butyrate and butanol limits the enhanced synthesis of butyl butyrate (Figure 1C) [23]. The synthesis of butyrate in *C. tyrobutyricum* L319 is achieved through the converting of butyryl-CoA and acetate with a butyryl-CoA/acyl-CoA transferase (*cat1*) (Figure 2A), which plays a vital role in the butyrate/acetate ratio [26]. In previous studies, it was shown that the heterologous expression of the aldehyde/alcohol dehydrogenase (*adhE2*) leads to a shortage of butyryl-CoA, hindering the pathway for the conversion of acetate to butyrate [19,32]. As a result, the strain CtADGBC exhibited a reduction in butanol production, and the different titer between butyrate and butanol was reduced (0.5 vs. 1.6), which was more favorable for the synthesis of butyl butyrate (Figure 2C). However, the titer of the byproduct, acetate, was still too high to escape. Further studies might be conducted to focus on reducing acetate production by silencing the related function genes. Furthermore, the reduced activity of the lipase, caused by suboptimal temperature and stressful conditions during fermentation, also contributes to the mismatch between precursors and products.

One of the advantages of *C. tyrobutyricum* as a cell factory is its wide range of substrates, particularly its natural advantages in utilizing biomass waste [10]. To our knowledge, *C. tyrobutyricum* could naturally use xylose as a sole carbon source, but it would grow worse than when feeding on glucose, which might be due to the short supply of extra energy for xylose utilization via the pentose–phosphate pathway (PPP) [29,33]. Previous studies have overcome the glucose-mediated carbon catabolite repression (CCR) by overexpressing the *xylT*, *xylA*, and *xylB* genes, which has significantly enhanced the uptake of xylose [34,35]. Furthermore, pretreating agri-food waste is necessary to release the inner sugars, a process that significantly increases costs and complicates the overall procedure [10]. Interestingly, we found that the strain CtADGBC could consume xylose much better when feeding on the SSP medium with no need for any genetic engineering disturbances on the uptake capacity and any chemical pretreatments for the wastes (Figure 6).

Moreover, aside from serving as a source for chitin extraction, there have been few reports on the utilization of shrimp shell waste as nitrogen sources for biorefinery [36]. The components of shrimp shells are mainly protein (20–40%), calcium carbonate (CaCO_3_, 20–50%) and chitin (15–40%), and other minor components such as lipids, astaxanthin [31]. Thus, the dissolution of calcium carbonate is a prerequisite for the release of protein and amino acids. Acid hydrolysis is a conventional and essential approach for processing shell waste. Considering the rich protein and amino acid concentrations in shrimp shell waste, and building upon our previous findings [37], *C. tyrobutyricum* becomes a promising host because of its high acid tolerance and capability of protein hydrolysis. Initially, when attempting to substitute the nitrogen source in the medium with shrimp shell powder, we directly cultured the strain with varying amounts of shrimp shell powder and glucose as components. However, we observed that the strain could not proliferate in this medium. Consequently, we consulted the ion formula in the CGM medium and supplemented it with additional ions. Subsequently, we identified the essential composition required to sustain the growth of the strain. Finally, we successfully developed a suitable fermentation medium for butyl butyrate production with xylose as the sole carbon source and shrimp shell waste as the sole nitrogen source, which would lower the cost by ~97% more than using commercial nitrogen sources during the fermentation (Table 2). Also, the cost would be further decreased due to the fact that xylose is abundant in the agricultural lignocellulosic biomass waste, which could be used as feedstock for fermentation. More importantly, we found that 5.9 g/L of butyl butyrate with a selectivity of 100% could be achieved in the strain CtADGBC using xylose and shrimp shell waste during batch fermentation in a 5 L bioreactor; the strain also achieved a xylose consumption rate of 1.3 g/L·h, which had never been achieved previously in *C. tyrobutyricum* when fed with xylose. This may be related to the fermentation environment containing xylose and shrimp shell waste, and is worth exploring further. In order to explore the influence of different batches and sources of shrimp shell powder on fermentation, we conducted repeated fermentation with different sources and batches of shrimp shell powder; the results were consistent, which excluded the influence of subsequent batches of shrimp shell powder on fermentation performance.

To further investigate the effects of xylose and SSP on the intracellular metabolism of strain CtADGBC, the transcriptome analysis was conducted to analyze gene transcription changes (Figure 7) (Appendix A). There is no clear study on the specific mechanism of xylose utilization in *C. tyrobutyricum*. By comparing the changes in the transcript levels of xylose-related genes under different xylose-metabolizing abilities, we found a significant up-regulation of the expression of *xylA* and *xylB* genes related to the xylose metabolism, which is a direct cause of the elevated xylose utilization ability. In addition, nitrogenases were all generally transcriptionally up-regulated in terms of nitrogen source utilization, which is the difference between SSP as a nitrogen source and other nitrogen sources during fermentation. The genes related to amino acid metabolism and transport were also transcriptionally up-regulated in the strains grown in the SSP medium because of the abundance of amino acids and the specific amino acid composition in the SSP [4]. In the present study, the potential of SSP as an alternative nitrogen source was evaluated to alter the xylose utilization capacity, and this finding about *C. tyrobutyricum* also requires further studies to reveal a more detailed and complete nitrogen source-mediated metabolic regulatory network.

## 5. Conclusions

In this study, a novel approach for butyl butyrate production was established by combining metabolic engineering and in situ esterification strategies. Then, the fermentation medium for butyl butyrate production was developed with xylose as the sole carbon source and SSP as the sole nitrogen source. The final generated strain CtADGBC could produce 5.9 g/L of butyl butyrate using xylose and SSP as feedstocks during batch fermentation in a 5 L bioreactor. Though the titer was much lower than that achieved in a recent study (62.59 g/L), it successfully reduced the cost by 97% in nitrogen sources and the high-selectivity production (100%) of butyl butyrate. To our knowledge, this is the first report about the utilization of xylose and SSP for butyl butyrate production. However, further studies will be conducted to enhance the titer of butyl butyrate by implementing an auto-regulated precursor-generating system utilizing novel genetic tools, such as CRISPRi/a. Additionally, the crucial role of the SSP medium in achieving high selectivity during ester synthesis will be thoroughly investigated.

## Figures and Tables

**Figure 1 foods-13-01009-f001:**
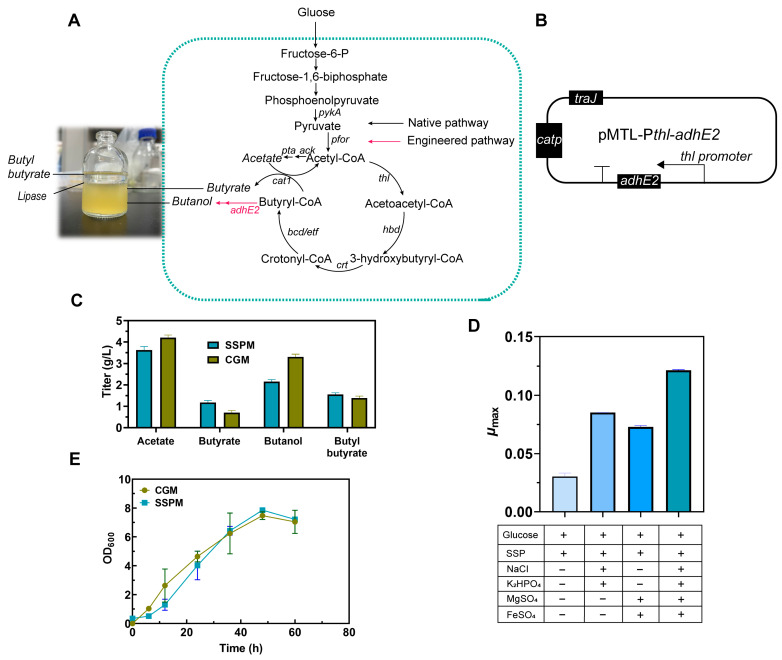
The biosynthetic pathway of butyl butyrate in *C. tyrobutyricum*. (**A**) Metabolic pathways in engineered *C. tyrobutyricum* for butyrate and butanol production from glucose. (**B**) The design of gene *adhE2* expression. (**C**) The main fermentation products of CtAD in CGM and SSP medium. (**D**) The optimization of the CGM. (**E**) The growth performance of CtAD in CGM and SSP medium. (Gene name and abbreviation: *pykA*: pyruvate kinase; *pfor*: pyruvate-ferredoxin/flavodoxin oxidoreductase; *ack*: acetate kinase; *pta*: phosphotransacetylase; *thl*: thiolase; *hbd*: beta-hydroxybutyryl-CoA dehydrogenase; *crt*: crotonase; *bcd*: butyryl-CoA dehydrogenase; *etf*: electron transferring flavoprotein; *cat1*: butyryl-CoA/acetate CoA transferase; *adhE2*: aldehyde/alcohol dehydrogenase).

**Figure 2 foods-13-01009-f002:**
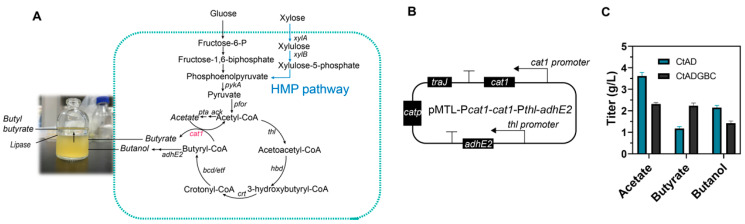
Balancing the ratio between butyrate and butanol by overexpressing *cat1* gene. (**A**) Metabolic pathways in engineered *C. tyrobutyricum* for butyrate and n-butanol production from glucose and xylose. (**B**) The design for *cat1* gene expression. (**C**) The main fermentation products of CtAD and CtADGBC in SSP medium. (Gene name and abbreviation: *xylA*: xylose isomerase; *xylB*: xylulokinase; HMP pathway: hexose monophosphate pathway).

**Figure 3 foods-13-01009-f003:**
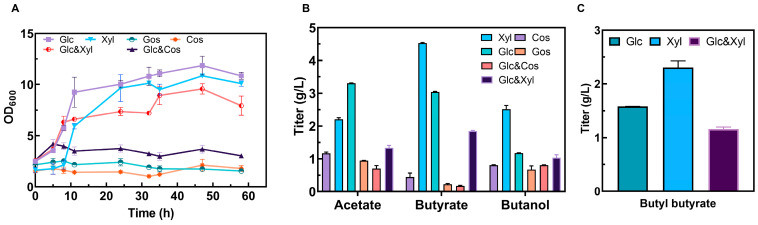
The utilization of various carbon sources by strain CtADGBC in SSP medium. (**A**) Growth performances using different carbon sources. (**B**) The titers of the main products during fermentation with different carbon sources. (**C**) The titers of butyl butyrate with glucose or xylose as carbon sources. The carbon mole (C-mole) of different carbon sources was maintained. Glc: glucose (30 g/L), Xyl: xylose (30 g/L), GOS: Galactooligosaccharides (5 g/L, MW~1200), COS: chitooligosaccharides (3 g/L, MW~2000), Glc & Cos: glucose and chitooligosaccharides (15 g/L and 1.5 g/L), Glc & Xyl: glucose and xylose (15 g/L and 15 g/L).

**Figure 4 foods-13-01009-f004:**
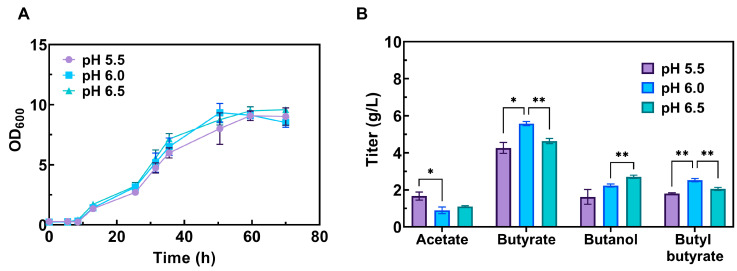
Impact of acidic conditions on the production of butyl butyrate. (**A**) Effects of different pH on the growth performance of CtADGBC. (**B**) Effects of different pH on the production of the main products. Results are presented as mean values ± standard deviation from at least two replicates (* *p* ≤ 0.05, ** *p* ≤ 0.01).

**Figure 5 foods-13-01009-f005:**
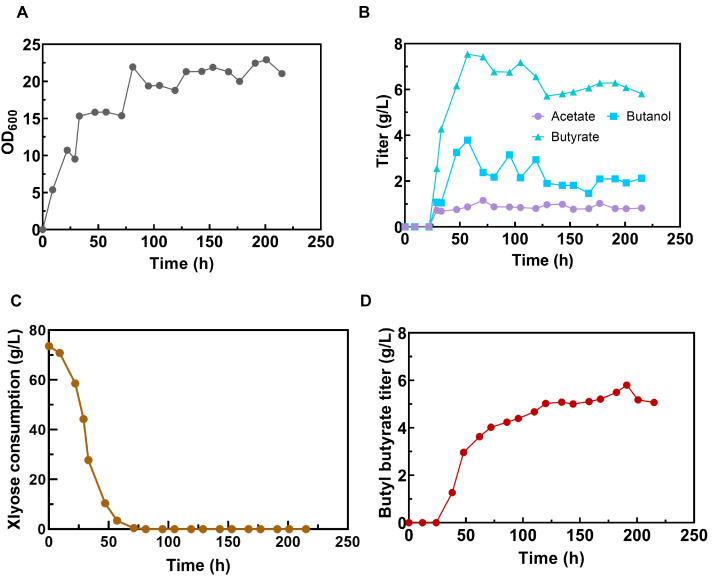
Batch fermentation of strain CtADGBC using xylose as sole carbon source and SSP as sole nitrogen source at 37 °C in a 5 L bioreactor. (**A**) Growth profile of CtADGBC. (**B**) Acids and alcohols in aqueous phase. (**C**) The consumption of xylose during the fermentation. (**D**) The production of butyl butyrate.

**Figure 6 foods-13-01009-f006:**
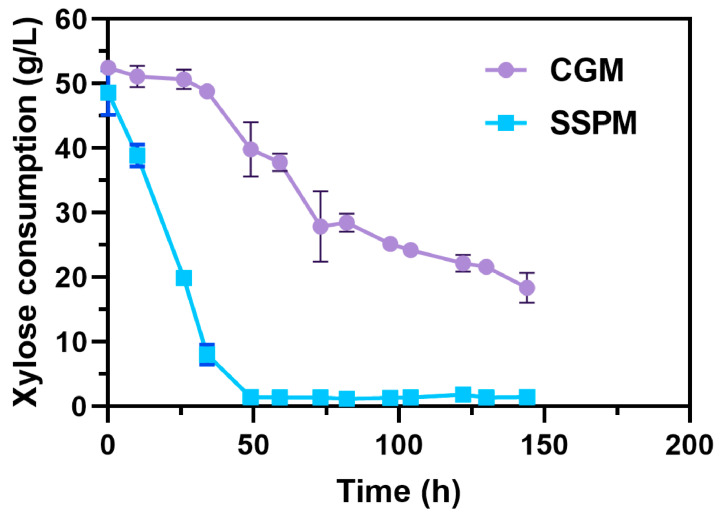
The differences in xylose consumption in CGM and the SSP medium.

**Figure 7 foods-13-01009-f007:**
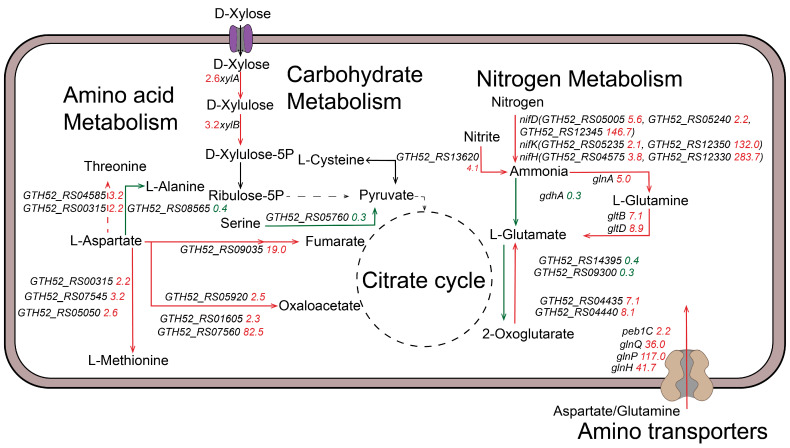
Significantly regulated genes of the CtADGBC strain, which were cultured in the SSP medium. The red arrows and numbers represent up-regulated genes and the corresponding fold changes, and the green ones represent down-regulated genes and the corresponding fold changes.

**Table 1 foods-13-01009-t001:** Strains and plasmids used in this study.

Strain/Plasmid	Description	Reference
	Plasmids	
pMTL82151	ColE ori, Cm^R^, pBP1 ori, TraJ	[25]
pMTL-P*thl-adhE2*	pMTL82151 derivate, *adhE2* gene under the control of the promoter *thl*	[27]
pMTL-P*thl-adhE2*::P*cat1*-*cat1*	pMTL82151 derivate, *adhE2* gene under the control of the promoter *thl* and *cat1* gene under the control of the natural promoter	This study
	Strains	
*C. tyrobutyricum*	L319	[24]
*E. coli* JM109	*E. coli,* for plasmid construction	Lab stock
*E. coli* CA434	*E. coli* HB101 with plasmid R702	[25]
CtAD	*C. tyrobutyricum* L319 derivate, harboring the plasmid pMTL-P*thl-adhE2*	This study
CtADGBC	*C. tyrobutyricum* L319 derivate, harboring P*thl-adhE2*::P*cat1*-*cat1*	This study

**Table 2 foods-13-01009-t002:** The economic evaluation of the nitrogen sources used in CGM and SSP media.

Medium	Nitrogen Source	Unit Price ($/t) **	Medium Costs ($/L) **	Butyl Butyrate Titer (g/L)	Costs of BB Titer ($/g) **
CGM medium	Commercial compounds *	~6000	0.0780	1.400	0.0560
SSP medium	Shrimp shell powder (SSP)	100–120	0.0025–0.0030	1.520	0.0016–0.0019

* Commercial compounds including tryptone, yeast extract and (NH_4_)_2_SO_4_. ** $: USD.

## Data Availability

The original contributions presented in the study are included in the article/Appendix A, further inquiries can be directed to the corresponding authors.

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
