# Peer review of "Metabolic and Bioprocess Engineering of Clostridium tyrobutyricum for Butyl Butyrate Production on Xylose and Shrimp Shell Waste"

_foods, 2024, doi:10.3390/foods13071009_

Round 1

Reviewer 1 Report

Comments and Suggestions for Authors

The manuscript is interesting, but some clarifications are necessary to improve the content.

1)      Abstract: The main objective is not clear. Was the objective to obtain a cheaper alternative medium using xylose and shrimp shells or was this possibility is a consequence of the modifications made to the strain? Or the goal was to produce butyl butyrate by engineering the strain?

2)      Line 124 and Figure 3: “…The carbon mole (C-mole) of different carbon sources were maintained.” But glucose concentration was 30  g/L and xylose, 25 g/L …How? Considering that the carbon sources used are of high purity, I do not see how the C - mole ratio was maintained. 30g/L of glucose does not have the same carbon content as 25 g/L of xylose. Please review.

3)      In the bioreactor, why the initial concentration of xylose was 70 g/L? The composition of SSP medium was the same as in the serum bottles? Explain in the discussion.

4)      Line 167: During the fermentation, samples were collected at different growth stages. In bottle experiments, are the analyzes followed the same sampling scheme at various fermentation times? Include this information in Figures 3B and 3C.

5)      Line 121: the catalytic synthesis of butyl butyrate was performed  using lipase Novozyme 435. When it was added in the serum bottles/ 5 L bioreactor? After the end of log phase? Did the conditions of the fermented medium favor the action of the enzyme?

6)       Line 283, sub-topic 3.5: “…butyl butyrate production by using xylose as the sole carbon source...”

Various carbon sources were tested in the experiments with the strain CtADGBC, but only in SSP medium. I understand that replacing the nitrogen source with shrimp shells results in a cheaper medium, but wouldn't the yield and productivity of butyl butyrate be greater in the original CGM medium containing xylose? Was this test carried out in a bioreactor?

7)      Line 328-330: Why did the synthesis of butyl butyrate not follow the generation of precursors (butyrate and butanol)? What should be considered? Was the enzyme action not efficient? Discuss this possibility.

8)       Line 397: “… the strain CtADGBC could consume the xylose much better when fed on SSP medium (Figure 6)..” Why ? Can this be attributed to changes in the strain or the simple presence of the shrimp shell? It is important to test CtADGBC using media containing other N sources.

9)       The best result reached 5.9 g/L of butyl butyrate, but other modifications in C. tyrobutyricum reached much higher values (62.59 g/L), see “  doi: 10.1016/j.ymben.2023.03.009. Epub 2023 Mar 21”. This must be commented  in the conclusion if the main objective of the study is to obtain butyl butyrate by making modifications to the strain.

Author Response

For research article

Response to Reviewer 1 Comments

1. Summary

2. Questions for General Evaluation

Reviewer’s Evaluation

Response and Revisions

Does the introduction provide sufficient background and include all relevant references?

Yes

Are all the cited references relevant to the research?

Yes

Is the research design appropriate?

Yes

Are the methods adequately described?

Can be improved

Are the results clearly presented?

Can be improved

Are the conclusions supported by the results?

Can be improved

3. Point-by-point response to Comments and Suggestions for Authors

Comments 1: Abstract: The main objective is not clear. Was the objective to obtain a cheaper alternative medium using xylose and shrimp shells or was this possibility is a consequence of the modifications made to the strain? Or the goal was to produce butyl butyrate by engineering the strain?

Response 1: Thank you for pointing this out. We agree with this comment. Hence, we have amended the clarification in the Abstract to explicitly state our objective, which is to identify a more cost-effective fermentation medium containing xylose capable of producing butyl butyrate with a selectivity of 100% in C. tyrobutyricum. The enhanced utilization of xylose in the SSP medium was then evidenced by the upregulation of relevant genes involved in ammonia synthesis, amino acid metabolism, and transport. While we did investigate the regulation of the ratio within the precursors to enhance the titer of the ester, achieving a higher titer necessitates further research in this area.

Comments 2: Line 124 and Figure 3: “…The carbon mole (C-mole) of different carbon sources were maintained.” But glucose concentration was 30 g/L and xylose, 25 g/L …How? Considering that the carbon sources used are of high purity, I do not see how the C - mole ratio was maintained. 30g/L of glucose does not have the same carbon content as 25 g/L of xylose. Please review.

Response 2: Agree. We have, accordingly, revised the calculation method of the C-mole in section 2.5 to emphasize this point. Furthermore, upon reviewing the xylose concentration data collected during the fermentation process in our source data, we identified a clerical error in the description of xylose C-mole. We have promptly rectified this issue in section 3.5: “For glucose (30 g/L) and xylose (30 g/L), their high purity enabled the calculation of corresponding concentrations using the molar amount of carbon formula. However, for chitooligosaccharide (3 g/L) and galactooligosaccharide (5 g/L), which are mixtures, precise determination via the formula is not feasible. Therefore, we calculated the corresponding concentrations according to the molar mass multiples.”.

The source data is provided below:

Comments 3: In the bioreactor, why the initial concentration of xylose was 70 g/L? The composition of SSP medium was the same as in the serum bottles? Explain in the discussion.

Response 3: Thank you for pointing this out. We will, accordingly, done an explanation to emphasize this point. Initially, a xylose concentration of 30 g/L was employed to assess different carbon sources in the serum bottles (see Figure 3), ultimately leading to complete consumption (data not shown). Subsequently, the xylose concentration was elevated to 50 g/L for comparison with other nitrogen sources in the CGM medium, also resulting in complete depletion by the end of the fermentation (see Figure 6). Therefore, in an effort to achieve higher titers consistent with the initial carbon source used in a previous study [1], we increased the xylose concentration to 70 g/L in the bioreactor, where it was likewise entirely utilized. Additionally, the composition of the SSP remained consistent with that used in the serum bottles.

1.      Guo, X., H. Zhang, J. Feng, L. Yang, K. Luo, H. Fu, and J. Wang. "De novo biosynthesis of butyl butyrate in engineered Clostridium tyrobutyricum." Metab Eng 77 (2023): 64-75.

Comments 4: Line 167: During the fermentation, samples were collected at different growth stages. In bottle experiments, are the analyzes followed the same sampling scheme at various fermentation times? Include this information in Figures 3B and 3C.

Response 4: Thank you for pointing this out. The data presented in Figures 3B and 3C correspond to the same samples depicted in Figure 3A. To facilitate a comparison of product composition throughout the fermentation process, we selected the final stage of the strain to assess fermentation performance across various carbon sources.

Comments 5: Line 121: the catalytic synthesis of butyl butyrate was performed using lipase Novozyme 435. When it was added in the serum bottles/ 5 L bioreactor? After the end of log phase? Did the conditions of the fermented medium favor the action of the enzyme?

Response 5: Thank you for pointing this out. We have, accordingly, revised the detail information to emphasize this point. (1) The adding of the lipase in the serum bottles/ bioreactors was carried out at the early of the log phase (~12 h). We have added the detail information in section 2.3, which was marked in red. (2) No, it didn’t. Considering the previous reports on Novozyme 435, the lipase was efficiently active at the temperature between 40 and 60 ℃. However, since the reactions were carried out during the growth of the strain, we chose the optimized temperature (37℃) for the strain rather than the reaction according to the previous report [1].

1.           Zhang, Z.-T., S. Taylor, and Y. Wang. "In situ esterification and extractive fermentation for butyl butyrate production with clostridium tyrobutyricum." Biotechnology and Bioengineering 114, no. 7 (2017): 1428-37.

Comments 6: Line 283, sub-topic 3.5: “…butyl butyrate production by using xylose as the sole carbon source...” Various carbon sources were tested in the experiments with the strain CtADGBC, but only in SSP medium. I understand that replacing the nitrogen source with shrimp shells results in a cheaper medium, but wouldn't the yield and productivity of butyl butyrate be greater in the original CGM medium containing xylose? Was this test carried out in a bioreactor?

Response 6: Thank you for pointing this out. We will, accordingly, done an explanation to emphasize this point. At the outset of the study, we observed the advantages of the SSP medium when supplemented with glucose (see Figure 1). The butyl butyrate titer achieved in the SSP medium was notably higher compared to that in the conventional CGM medium when glucose was utilized as the carbon source. Moreover, an increase in butyl butyrate titer was evident in the SSP medium when supplemented with xylose. Consequently, the SSP medium supplemented with xylose was deemed the most favorable composition for further investigation in the 5 L bioreactor.

Comments 7: Line 328-330: Why did the synthesis of butyl butyrate not follow the generation of precursors (butyrate and butanol)? What should be considered? Was the enzyme action not efficient? Discuss this possibility.

Response 7: Thank you for pointing this out. We have, accordingly, revised in Discussion (paragraph one, marked in red) to emphasize this point. The rationale behind our hypothesis lies in the low activity of the lipase. Given that the synthesis took place in situ, the acidic and solvent conditions present during fermentation may have led to a reduction in lipase activity. Furthermore, suboptimal temperature conditions for the lipase may have contributed to its reduced activity.

Comments 8:  Line 397: “… the strain CtADGBC could consume the xylose much better when fed on SSP medium (Figure 6)..” Why ? Can this be attributed to changes in the strain or the simple presence of the shrimp shell? It is important to test CtADGBC using media containing other N sources.

Response 8: Thank you for pointing this out. We are going to, accordingly, done an explanation to emphasize this point. The xylose consumption was found enhanced in SSP medium compared with CGM (commercial N sources) when using strain CtADGBC (Figure 6). The difference between these two media was only the N sources. Thus, it might be the SSP medium that enhanced the xylose consumption alone.

Comments 9: The best result reached 5.9 g/L of butyl butyrate, but other modifications in C. tyrobutyricum reached much higher values (62.59 g/L), see “doi: 10.1016/j.ymben.2023.03.009. Epub 2023 Mar 21”. This must be commented in the conclusion if the main objective of the study is to obtain butyl butyrate by making modifications to the strain.

Response 9: Agree. We have, accordingly, revised the comment in the conclusion (marked in red) to emphasize this point. As discussed in Conclusion, our novel fermentation medium incorporating SSP demonstrated a remarkable 97% reduction in the cost of nitrogen source. This study represents the pioneering utilization of shrimp shell waste for butyl butyrate synthesis. Further investigations are warranted to enhance butyl butyrate titers; however, we observed significant improvement in xylose utilization within the SSP medium, providing valuable insights for future strain modifications.

4. Response to Comments on the Quality of English Language

Point 1:

None

5. Additional clarifications

None

Reviewer 2 Report

Comments and Suggestions for Authors

In the paper entitled "Metabolic and bioprocess engineering of Clostridium tyrobutyricum for butyl butyrate production on xylose and shrimp shell wastes," a butanol synthetic pathway was constructed in C. tyrobutyricum, and then efficient butyl butyrate production through in situ esterification was achieved by the supplementation of lipase into the fermentation. The butyryl-CoA/acyl-CoA transferase (cat1) was overexpressed to balance the ratio between precursors butyrate and butanol. Then, a suitable fermentation medium for butyl butyrate production were obtained with xylose as the sole carbon source and shrimp shell wastes as the sole nitrogen source. This innovative approach could save production cost (~97%) and opens up possibilities for converting agri-food waste into other high-value products.

However, I have the following comments:

1) Please discuss a few demerits of using Clostridium tyrobutyricum for the microbial conversion of agri-food wastes.

2) Please delve into more detail for key findings regarding the fermentation medium optimization.

3) Figure 7 should be made more readable.

4) Please elaborate conclusion more by adding future work.

Author Response

For research article

Response to Reviewer 2 Comments

1. Summary

2. Questions for General Evaluation

Reviewer’s Evaluation

Response and Revisions

Does the introduction provide sufficient background and include all relevant references?

Can be improved

Are all the cited references relevant to the research?

Can be improved

Is the research design appropriate?

Can be improved

Are the methods adequately described?

Can be improved

Are the results clearly presented?

Can be improved

Are the conclusions supported by the results?

Can be improved

3. Point-by-point response to Comments and Suggestions for Authors

Comments 1: Please discuss a few demerits of using Clostridium tyrobutyricum for the microbial conversion of agri-food wastes.

Response 1: Thank you for pointing this out. We agree with this comment. Therefore, we have supplemented the clarification about this part in section Discussion, Paragraph two, which was marked in red. To our knowledge, the application of C. tyrobutyricum was hindered for two aspects. Since glucose and xylose are the most common sugars released from agri-food wastes, the first one is the poor utilization of the xylose, especially the using of mixture of the glucose and xylose, which we have mentioned in section discussion, paragraph two. The other one is the need of the pretreatments during the fermentation on the agri-food wastes. The pretreatments of the agri-food wastes are required to release the inner sugars, which totally increase the cost and complicates the process. So, we found that the strain CtADGBC could consume the xylose much better when fed on SSP medium with no need of any genetic engineering disturbances on the uptake capacity and any chemical pretreatments for the wastes.

Comments 2: Please delve into more detail for key findings regarding the fermentation medium optimization.

Response 2: Agree. We have, accordingly, detailed the finding of the SSP medium in the section Discussion to emphasize this point. As is: Initially, when attempting to substitute the nitrogen source in the medium with shrimp shell powder, we directly cultured the strain with varying amounts of shrimp shell powder and glucose as components. However, we observed that the strain could not proliferate in this medium. Consequently, we consulted the ion formula in the CGM medium and supplemented it with additional ions. Subsequently, we identified the essential composition required to sustain the growth of the strain.

Comments 3: Figure 7 should be made more readable.

Response 3: Agree. We have, accordingly, made the figure more readable by adjusting the front size. The revised figure was posted as follows:

Comments 4: Please elaborate conclusion more by adding future work.

Response 4: Thank you for pointing this out. We agree with this comment. Therefore, we have, accordingly, added future work that might be done to further this study. The future work will be carried out in two parts. The first part is to increase the final titer of the butyl butyrate using the novel methods, such as CRISPRi system, to dynamically regulate the ratio of precursors butyrate and butanol to achieve efficient synthesis of esters. The second part is to further explored the specific regulation of ester synthesis selectivity by shrimp shell powder medium, and some transcriptional regulators that may be involved in this need to be further discovered.

4. Response to Comments on the Quality of English Language

Point 1:

None

5. Additional clarifications

None

Reviewer 3 Report

Comments and Suggestions for Authors

Dear Authors,

The paper "Metabolic and bioprocess engineering of Clostridium tyrobutyricum for butyl butyrate production on xylose and shrimp shell wastes" is a great example of how genetic engineering can be used to create a fermentation process that can use an industrial waste (shrimp shell) to make butyl butyrate and butanol, which are useful chemicals for industry.

The merits of the work include the clarity of the presentation and the presentation of the results.

Our suggestion is for the authors to correct some typographical errors and provide more detailed explanations of some of the experimental procedures carried out.

Please correcting some of these minor errors.

Congratulations!

Regards,

Reviewer

Author Response

For research article

Response to Reviewer 3 Comments

1. Summary

Thank you very much for taking the time to review this manuscript. Please find the detailed responses below and the corresponding revisions highlighted changes in the re-submitted files.

2. Questions for General Evaluation

Reviewer’s Evaluation

Response and Revisions

Does the introduction provide sufficient background and include all relevant references?

Yes

Are all the cited references relevant to the research?

Yes

Is the research design appropriate?

Yes

Are the methods adequately described?

Yes

Are the results clearly presented?

Yes

Are the conclusions supported by the results?

Yes

3. Point-by-point response to Comments and Suggestions for Authors

Comments 1: Our suggestion is for the authors to correct some typographical errors and provide more detailed explanations of some of the experimental procedures carried out.

Response 1: Thank you for pointing this out. We agree with this comment. Therefore, we have carefully checked those errors and made them corrected, the changes were marked in red in the revised manuscript. Besides, some detail information has been added in the right place, which has been also marked in red.

4. Response to Comments on the Quality of English Language

Point 1:

None.

5. Additional clarifications

None.

Round 2

Reviewer 2 Report

Comments and Suggestions for Authors

Accept in present form.